# Impact Evaluation of Score Classes and Annotation Regions in Deep Learning-Based Dairy Cow Body Condition Prediction

**DOI:** 10.3390/ani13020194

**Published:** 2023-01-04

**Authors:** Sára Ágnes Nagy, Oz Kilim, István Csabai, György Gábor, Norbert Solymosi

**Affiliations:** 1Centre for Bioinformatics, University of Veterinary Medicine, 1078 Budapest, Hungary; 2Department of Physics of Complex Systems, Eötvös Loránd University, 1117 Budapest, Hungary; 3Androvet Ltd., 1182 Budapest, Hungary

**Keywords:** deep learning, dairy cow, body score, prediction, accuracy

## Abstract

**Simple Summary:**

The body condition of dairy cattle is an essential indicator of the energy supply of the animals. Various scoring systems are used in practice to quantify body condition. These systems rely on visual observation of different body parts and sometimes on collecting tactile data. In all cases, scoring requires expert knowledge and practice and is time-consuming. Therefore, it is rarely carried out on livestock farms. However, for animal husbandry and veterinary practice, it would be meaningful to have data on the condition of the animals continuously or even daily, which is not feasible with expert scoring. We investigated how computer vision-based supervised deep learning, specifically neural networks, can automate body condition scoring. To execute this, we have used video recordings of the rumps of a large number of animals. We have trained and tested various convolutional neural networks with this collected data. Scoring by trained networks yielded results that met or exceeded the agreement among experts. We have made our trained neural networks freely available, using these as pretrained models. Those working on similar developments can achieve even better results with less data collection required with their own fine-tuning.

**Abstract:**

Body condition scoring is a simple method to estimate the energy supply of dairy cattle. Our study aims to investigate the accuracy with which supervised machine learning, specifically a deep convolutional neural network (CNN), can be used to retrieve body condition score (BCS) classes estimated by an expert. We recorded images of animals’ rumps in three large-scale farms using a simple action camera. The images were annotated with classes and three different-sized bounding boxes by an expert. A CNN pretrained model was fine-tuned on 12 and 3 BCS classes. Training in 12 classes with a 0 error range, the Cohen’s kappa value yielded minimal agreement between the model predictions and ground truth. Allowing an error range of 0.25, we obtained minimum or weak agreement. With an error range of 0.5, we had strong or almost perfect agreement. The kappa values for the approach trained on three classes show that we can classify all animals into BCS categories with at least moderate agreement. Furthermore, CNNs trained on 3 BCS classes showed a remarkably higher proportion of strong agreement than those trained in 12 classes. The prediction precision when training with various annotation region sizes showed no meaningful differences. The weights of our trained CNNs are freely available, supporting similar works.

## 1. Introduction

Body condition scoring of cattle is a widespread, noninvasive and easy-to-use but subjective and time-consuming method for estimating an animal’s saturation of subcutaneous fat deposits [1,2]. The different scoring systems infer the animal’s energy supply from the coverage of the lumbar, pelvic and tail head regions [2]. The accumulation of body stores is quantified using a numerical scale, where lean individuals have low values and overweight individuals have high values [3]. The saturation of fat and energy stores can provide important guidance for farm management. Several studies have shown how a shift in body condition score (BCS) away from what is ideal is associated with changes in production [4]. Condition scores that are too high or low may indicate several health issues or mismanagement [4]. The association between a high condition score and the risk of ketosis is well known [4,5]. Furthermore, it is also associated with other metabolic problems (e.g., fatty liver) [6] or placenta retention [5]. Too low a condition score can be associated with lameness and reduced milk production [4]. Several pathological processes correlate with a decrease in BCS (e.g., metritis, inactive ovaries, displaced abomasum or more days open) [4]. Since pathological changes are often closely related to changes in BCS [4] rather than a specific condition score, it is understandable that continuous and reliable herd-level condition scoring would be an essential aid to dairy herd management [7,8,9,10,11]. This is hampered by the fact that scoring requires trained staff [2], and herd-level scoring is time-consuming. The volatility of intraobserver and interobserver agreements makes the BCS data generated challenging to use [12]. Mullis et al. [13] showed that the agreement between two experts’ BCS values is moderate, while Song et al. [14] found that inter- and intraobserver agreement is weak and moderate, respectively.

Our study aims to investigate the accuracy with which supervised machine learning, specifically deep convolutional neural network (CNN)-based Detectron2 models, can be used to recover the BCS classes estimated by an expert using images of cows taken with a simple RGB camera. As a first approach, we investigate the quality of the predictions of the CNNs trained on our 12-level BCS scoring. In the following approach, we investigate the quality of the predictions of CNNs trained on three BCS classes corresponding to four different target intervals: T1 for calving (days in milk (DIM): 0) with dry period (DIM from −60 to −1) and dry off period (DIM > 300 and DIM < −60), T2 for early (DIM: 1–30) and mid-lactation (DIM: 101–200), T3 for peak lactation (DIM: 31–100) and T4 for late lactation (DIM: 201–300). Furthermore, we study how different region of interest (ROI) rectangles of the rump cause variation in the predictive power of our models.

## 2. Materials and Methods

### 2.1. Data Collection

Digital video recordings were taken over the course of 2 years with an SJCAM 4000 RGB camera at three large-scale dairy cattle farms in Hungary (farm F1: 1150 cows; F2: 880 cows; F3: 960 cows), with the camera positioned in the rotary milking parlor pointing at the animals’ rumps. Not all of the milking cows were filmed in each video. To avoid overrepresentation of any cow with a given body scoring in the data set, we skipped at least a month between any two videos taken at a single site to ensure the conditions of the animals had changed.

### 2.2. Data Preprocessing

The recorded videos were annotated later by an expert remotely using the Visual Object Tagging Tool (VoTT, v2.2.0) [15], assigning both a bounding rectangle to the animal’s rump as well as an estimated BCS. Scoring was performed in the range of 1–5. In this range, 12 levels were defined: 1–2.5 and 4–5 were divided into 0.5 score intervals, while 2.5–4.00 was divided more finely with 0.25 score intervals [2]. The scoring was performed continuously with movie footage, drawing the bounding boxes and making the BCS inference from only the image captured where the rump was at the closest point to the camera. These were the final images used to build the training, validation and test sets. After annotating the videos produced in the study, the same expert rechecked and adjusted the annotations with Label Studio [16]. The F1 and F2 farm images were split into the training and validation sets. This was performed with stratification of the BCS scores. By randomly selecting 80% of the images within each score level for the train set, we created the validation set with the remaining pictures. This ensured that the training and validation sets contained the same class distributions, thereby making the validation set loss an appropriate metric for model choice decisions. The annotated images from site F3 were retained as an independent test set. The number of images per score in each set is summarized in Table 1. Three size bounding boxes were generated for each image automatically from the initial annotations (See Appendix A: Automated 3-box size annotation).

### 2.3. Choice of Model Architecture

Body scoring is an object detection and classification problem. For this reason, we chose an object detection model that could leverage both bounding box and class ground truth annotations. The Faster R-CNN architecture [17] has a shared model internal representation for the joint localization and classification prediction tasks. This joint learning task is apparent by inspecting the form of the loss function used for training $\ell =\ell_{cls}+\ell_{bbox}$. Due to the state-of-the-art results, the Detectron2 [18] implementation of the Faster R-CNN architecture was chosen. All models were downloaded from the Model Zoo code repository with network parameters pretrained on the COCO Dataset.

### 2.4. Evaluation Metrics

The performance of our model’s localization could be reviewed in terms of the bounding box prediction average precision at IoU = 0.50 (AP50). This value increases when the predicted bounding boxes overlap more with the ground truth annotation bounding boxes. In addition to detecting an object in the image, Detectron2 estimates a class probability distribution over all classes. As the final classification, it assigns the object to the class for which the probability is the highest among all possible classes. The quality of the predictions was quantified using Cohen’s kappa [19] and accuracy. Kappa is defined as $kappa=\left(P_{0}-P_{e}\right)/\left(1-P_{e}\right)$, where $P_{0}$ is the observed agreement between the ground truth and predicted classes and $P_{e}$ is the probability change agreement between the model prediction and annotation ground truth. Cohen’s kappa values can be interpreted as follows: 0–0.20 = no, 0.21–0.39 = minimal, 0.40–0.59 = weak, 0.60–0.79 = moderate, 0.80–0.90 = strong and above 0.90 = almost perfect agreement [20]. Following the “one-versus-all” scheme, the accuracy was calculated by comparing each class to the remaining levels with the formula accuracy=(TP+TN)/(TP+TN+FP+FN), and from these results, the overall accuracy was reported as the mean.

### 2.5. Model Screening

To pre-screen for the most appropriate pretrained model for the BCS task, 10 pretrained models of Detectron2 [18] were further trained and validated on our data sets, each with identical hyperparameters for 15 epochs. In this model selection phase, the raw BCS scores were classified into three classes (BCS classes: <2.5, between 2.5 and 3.75 and >3.75). The R_50_FPN_3x model gave the lowest validation loss, so we selected and used this pretrained model in all further experiments (Table 2).

### 2.6. Model Training and Prediction

#### 2.6.1. With 12 BCS Classes

Using the selected pretrained model, the first experiment was run by training, validating and testing of the 12-level ordinal scoring annotations. This was repeated for each of the three bounding box sizes (see Figure 1) separately. Using the model’s optimal weights at checkpoints with the lowest validation losses and AP50s, we made predictions for the validation and test sets. The class where the probability distribution (output by the model) had its maximum was taken as the estimated body score. We also kept the “predicted class probabilities”, as they are a measure of model confidence for a given prediction.

Following the approach of Yukun et al. (2019) [21], we evaluated the model predictions with 0, 0.25 and 0.5 error thresholds to account for the ordinal nature of the body condition scores and thereby allowed “near miss” predictions to be defined as correct, reducing metric stringency.

#### 2.6.2. With Three BCS Classes for Four Practical Target Intervals

In addition to the 12-point BCS annotation, we also assessed the quality of the predictions according to more broad BCS classes of practical relevance. Different BCS ranges are considered optimal at different stages of lactation. These target intervals are summarized in Table 3. We relabeled all the original BCSs according to four different threshold regime (T1, T2, T3 and T4) target ranges: T1, the optimal target interval for calving (days in milk (DIM): 0) and the dry period (DIM from −60 to −1) and dry off period (DIM > 300 and DIM: < −60); T2 for early (DIM: 1–30) and mid-lactation (DIM: 101–200); T3 for peak lactation (DIM: 31–100) and T4 for late lactation (DIM: 201–300). We created three new BCS classes for each type of thresholding: below the target interval, the target interval and above the target interval. The training, validating and testing sessions were re-run with these three new classes for each of the four thresholding regimes and for each of the three bounding box sizes. Each experiment used the same input images, where the label definitions varied between each. In Table 3, the linked figure shows the proportion of images after reclassification in the training, validation and test sets. The proportion of classes was the same in the training abd validation set (farm F1 and F2) and the test (farm F3) set. This allowed for fair testing, as the class imbalance was the same in training and validation and test sets.

To compare the training process between training on 3 classes and training on 12 classes, we repeated the analysis in a way where 12 classes were predicted, but they were then reclassified to the new 3 classes at the inference time (see Appendix A).

## 3. Results

### 3.1. Prediction with 12 BCS Classes

The quality of the predictions for the 12 classes, based on the training with 12 classes, is shown in Figure 2. The *x* axis represents the results thresholded by predicted class probabilities. As the threshold increased, only images classified with high model confidence were used to calculate kappa and the accuracy. Traversing the *x* axis of each plot gives an idea of how the data clustered near the learned decision boundary in high dimensional space. Images near the decision boundary had a low predicted class probability (model confidence). Images with high confidence were to be classified better.

When training and evaluating in 12 classes and allowing for an error range of 0, the kappa value on the test set yielded minimal agreement but worse agreement above a class prediction probability of about 75%. For the validation set, the agreement was similar to below 75% class prediction probability. In contrast, above this level, the agreement was weak for the l and s boxes and even weaker than the minimum for the m box. If we allowed an error range of 0.25, then we obtained minimum agreement below the 50% class prediction probability on both the test and the validation set, and above that, the maps fell into the weak agreement range. When allowing an error range of 0.5, we had curves running above or close to the strong cut point with a class prediction probability of about 60% on both the test and validation sets. However, from 65% to 70%, we found almost perfect agreement.

### 3.2. Prediction with Three BCS Classes

Figure 3 shows the quality of the predictions made on only 3 BCS classes instead of 12 BCS classes. The three classes were created according to the four target ranges (T1–T4) presented in Table 3. The expert’s scores assigned to the images were reassigned to one of the three BCS classes and then used to perform the training, with the prediction also being made for three classes. In this three-class regime, the network made high-quality predictions. This type of classifier may be more relevant for farms to give lower-resolution but reliable predictions and act more as a screening tool. If animals are predicted to be out of the target, then they can be further investigated.

In a second “control” approach, we trained the network on 12 BCS classes and returned 12 classes with the predictors as in Figure 2. We then reclassified the 12 classes into the 3 BCS classes. This was also repeated for each T1–4 regime. These results can be seen in Appendix A. The traversal of model confidence thresholding showed more noise. This is understandable, as the body scoring classes are ordinal but not continuous [14]. To further compare the two approaches, we examined the difference in the kappa and accuracy values for the joint class classification probabilities. For this, we subtracted the values of the trained set on 12 classes from the values of the trained set on 3 classes. In Appendix A, the mean and the standard deviation of the differences are plotted as outlined in the Appendix A. The methods showed similar performances.

The differences between the prediction goodness of annotation boxes obtained from the networks taught by the three classes are summarized in Appendix A. We may observe the same trends for accuracy and kappa for all target ranges in the box prediction differences. It can also be seen that the differences between the boxes were an order of magnitude smaller than those obtained from the networks taught by classes of 12 and 3. Based on the test set, the prediction precision per box in descending order for the target ranges was as follows: T1: m, s, l; T2: s, m, l; T3: l, s, m; T4: m, l, s. In the validation set, the prediction precision per box in descending order for the target ranges was as follows: T1: m, l, s; T2: m, l, s; T3: s, l, m; T4: m, l, s.

As we traversed the *x* axis for each result, we had fewer and fewer predicted images to evaluate the metric. Figure 4 summarizes the proportion of the initial (test or validation) data set corresponding to each predicted class probability value. For the neural networks trained on 12 classes, at a predicted class probability value of 40%, half of the images were already dropped. However, for the networks taught by 3 classes, even at the highest predicted class probability value, half of the images or slightly less than half of the images were still part of the analysis. The three-class classification could be considered more of an “easy” task for the network, so higher model confidence for the predicted classes was reasonable. We found variation between the results of the three box sizes to not be very large, indicating that the supervised signal was mainly found within the medium box area.

Regardless of the target range (T1–T4), the kappa values of the approach trained and evaluated in the three classes showed that we could classify all animals into BCS categories with at least moderate agreement. Figure 5 shows the proportion of predictions based on training for different target ranges with different box sizes that resulted in at least strong agreement.

## 4. Discussion

Several possible approaches to estimating cattle body conditions using neural networks exist [21,22,23]. The more common approach in the literature is to use recordings of animals scored with high-resolution scoring, such as in 0.25 or 0.5 unit increments, to train the neural network and to test how the trained network performs in terms of prediction reliability at the same scale. Assume no discrepancy is allowed between the observed and predicted scores. In that case, these approaches yield weak agreement, as indicated by the kappa=0.45 value of Yukun et al. [21]. If we allowed some variation between the observed and predicted scores, both the kappa and accuracy values improved. As the first step in our investigation, we followed the approach, and similar results were obtained around the reliability values presented by other authors [21,22,24,25,26,27,28].

However, in addition to these high-resolution score classes, we felt it was worthwhile to investigate the prediction quality that could be achieved for practically important condition score classes. We conducted this investigation in two forms. In one, we trained a neural network on the high-resolution, detailed 12-point level data set, and then both the predictions and original expert scores were classified into three condition categories. Thus, there were a target range category (T1, T2, T3 and T4) and categories below and above the target range. In the second approach, we trained the neural network with the data set already divided into three categories and made predictions corresponding to the three classes. The test and validation set’s predictions showed that the latter approach gave better results with lower noise. However, after splitting into three categories, the networks trained on the 12 categories also gave good results, but these results were noisier and could be evaluated to be a less robust approach.

We also considered the probability that the neural network assigned a BCS class to the object detected in each image. The kappa and accuracy values presented were evaluated as a function of this class assignment probability. It can be seen that the precision of the predictions improved with an increasing class assignment probability. Nevertheless, as the class ranking probability increased, fewer images could be considered in evaluating the reliability of the predictions. When comparing the networks trained on 12 classes and 3 classes, it can be seen that the class assignment probabilities were lower for the former than for the latter. For this reason, as we increased the threshold of the classification probability of the images included in the prediction precision analysis, the number of usable images predicted by the neural network trained on the 12 classes decreased rapidly, while for the networks trained on 3 classes, the number of usable images decreased much more slowly as the classification probability threshold increased. Even at the highest threshold, roughly half of the images were retained. This approach has not been found in the literature, where the prediction precision was analyzed in conjunction with the classification probability. However, the results show that it significantly affects the prediction quality and thus the practical efficacy of body condition prediction based on neural networks. Nevertheless, it is still problematic that the number of images with a higher classification probability was less than the number of animals in the set used for prediction. The results show that at the most reliable classification probability threshold, we lost half of the animals, which means we obtained reliable conditioning information for half of the animals on a given day. However, we aim to obtain daily information on all individuals in a given herd. We see several possibilities to address this problem, which further studies could clarify. One approach could be based on the fact that animals are snapped not only once but several times during milking in the carousel systems. We could identify the highest probability class from their class distribution if we predicted each of these. Nevertheless, we can conclude from Figure 5 that the prediction of CNNs trained in class 3 showed a significant proportion of strong agreement which was better than the inter-rater agreement found in the literature.

The use of practical thresholds in the 3-grade approach trained in 12 classes and assessed in 3 classes is problematic. Several authors showed that high prediction reliability can be obtained in error ranges of 0.25 or 0.5 [21,22,24,25,26,27,28]. However, when we tried to assign this to the practical condition categories we used, it was impossible for many to decide which practical condition interval to place an animal in with an error range of 0.25 or 0.5. It is important to emphasize that our study aimed to investigate the reliability with which a neural network can reproduce the scores and score categories of animals scored by an expert. It is also possible to construct a ground of truth from the scores of several experts rather than one. However, it is worth considering that the agreement between the scores of two independent experts is weak or moderate based on the literature. The results reported by Mullis et al. [13] show that Cohen’s kappa of the agreement between two experts’ BCS values is 0.62 and 0.66, while Song et al. [14] found that an inter-assessor agreement kappa = 0.48, while the intra-assessor agreement kappa is 0.52 and 0.72. Thus, the prediction precision of a neural network built on this basis could easily be worse, not better.

In our study, we used a test set from an independent site in addition to the validation set to see the robustness of the neural network prediction. After all, it was expected that if animals from the same farm were given the training and validation sets, the prediction for the validation set would be better than the predictions for an utterly independent farm. Surprisingly, the predictions of the networks trained on the three classes differed little for the test and validation sets. A further interesting feature of our results is that the prediction precision based on the three types of annotation boxes (large, medium and small) also differed very little. We chose three different-sized annotation boxes because we might have thought a large box contained more information than the algorithm could capture. Conversely, it seems that a medium-sized box contained the most usable information, which still had little noise. Next in line was the small box, which gave slightly better results than the large box. Here, we can think of it as containing less information and less noise. In contrast, the big one had more noise with more information. Thus, the order was that the middle one came after the small box, followed by the large box in terms of prediction performance.

In our work, we deliberately did not use complementary tactile examinations in the generation of expert scores because when teaching a neural network based purely on images, this information is not available to the algorithm, so the expert’s information in scoring is richer than what we can offer the neural network.

The results show that the quality of training and prediction from two-dimensional images taken with a simple sports camera using Detectron2 is not inferior to the prediction results based on three-dimensional cameras or on scoring with tactile detection [29,30]. An additional option to consider to improve the prediction quality could be to use ensemble prediction of different trained networks as the final output.

The results for the weights generated for the CNNS are publicly available. Others can use this as a pretrained model for training neural networks on similar images. Thus, presumably, they can create their neural networks while using fewer images to predict BCS categories. The presented results suggest that similar outcomes can be expected in condition scoring for other breeds and types of utilization. However, this assumes that similar CNNs should be trained on data sets generated by a scoring system applied to given breeds and types of utilization. Thus, the trained algorithm presented here cannot be used one-to-one for other breeds and utilization types.

## 5. Conclusions

Our results conclude that CNN training on classes corresponding to practically relevant target ranges gives more robust and precise predictions than training on high-resolution classes. With predictions based on target interval training, we obtained similar or even better results than the agreement between experts. The prediction precision based on training with various annotation regions showed no meaningful differences.

## Figures and Tables

**Figure 1 animals-13-00194-f001:**
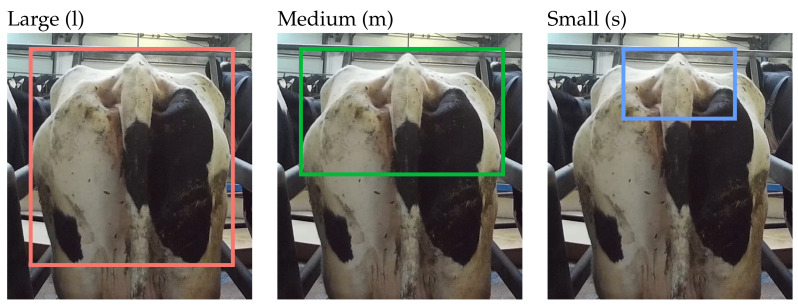
Annotation rectangles. The large (l) box was placed across the width of the two tuber coxae from the anterior coccygeal vertebrae (tail head) to mid-thigh. The medium (m) box was the entire width between the coxal tuberosities from the tail head to the symphysis pelvis. The small (s) box only framed the ischial tuberosities and the tail head, containing the depression between the tuber ischii and the anterior coccygeal vertebrae, and the area between the tuber coxae and the sacral spinous processes.

**Figure 2 animals-13-00194-f002:**
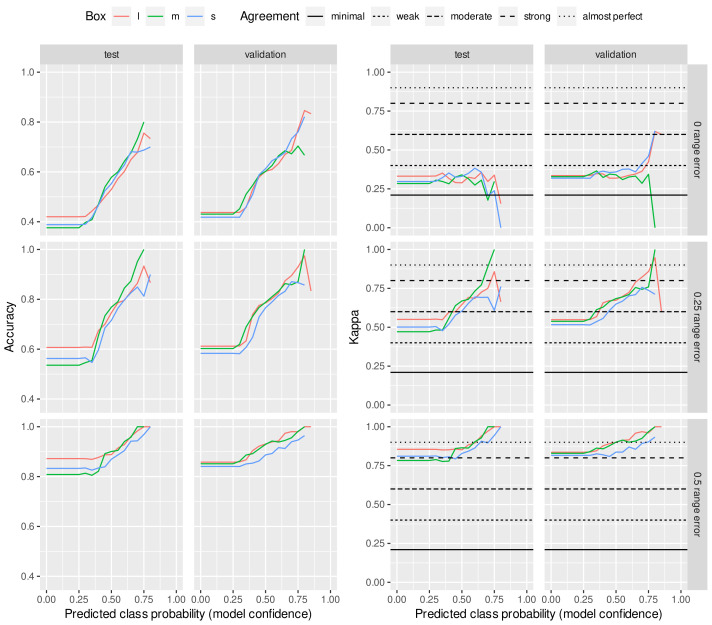
Model trained and evaluated with 12 BCS classes. Prediction confidence values (accuracy and Cohen’s kappa) were estimated on test and validation sets as a function of predicted class probability. Three error ranges (0, 0.25 and 0.5) were allowed for agreement analysis of the expert-given and predicted scores. The horizontal lines represent the thresholds of McHugh [20] to interpret Cohen’s kappa values.

**Figure 3 animals-13-00194-f003:**
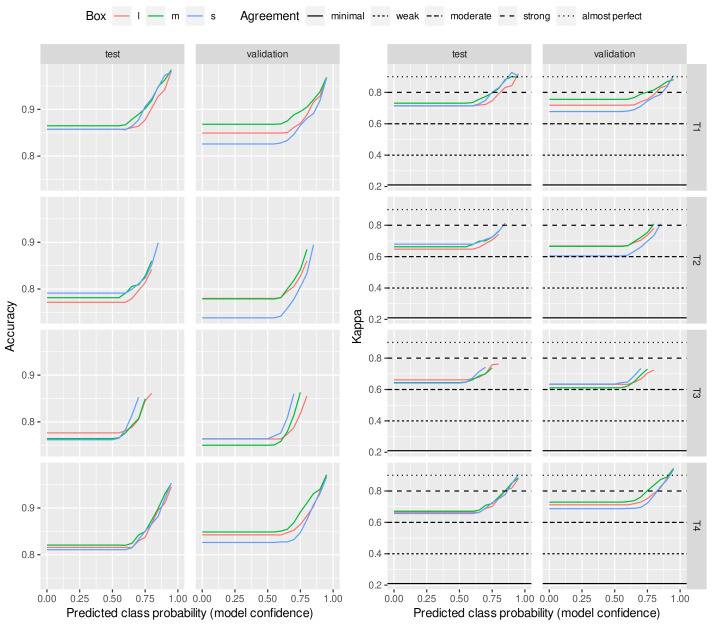
Model trained and evaluated with three BCS classes. Prediction confidence values (accuracy and Cohen’s kappa) were estimated on test and validation sets as a function of predicted class probability. The horizontal lines represent the thresholds of McHugh [20] to interpret Cohen’s kappa values.

**Figure 4 animals-13-00194-f004:**
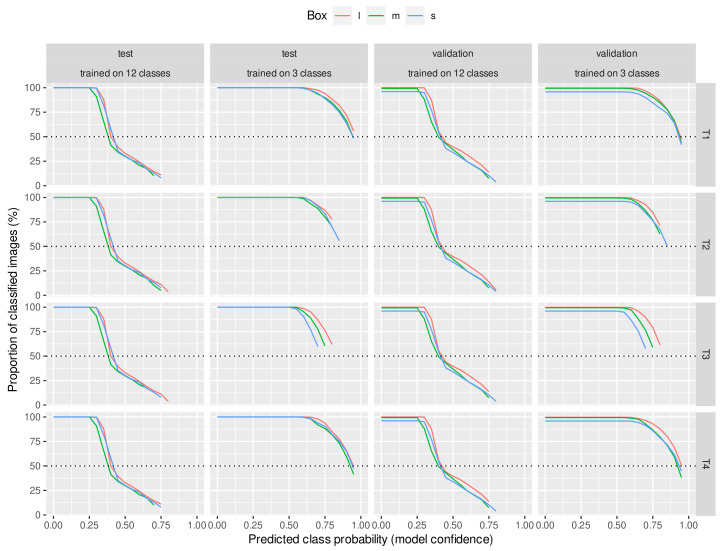
Proportion of classified images. As the threshold cutoff is more stringent, the number of images left gets smaller. The reduction occurred to a larger extent for the 12 class experiments. This gives an insight into the distribution of images concerning the learned decision boundary in the model representation.

**Figure 5 animals-13-00194-f005:**
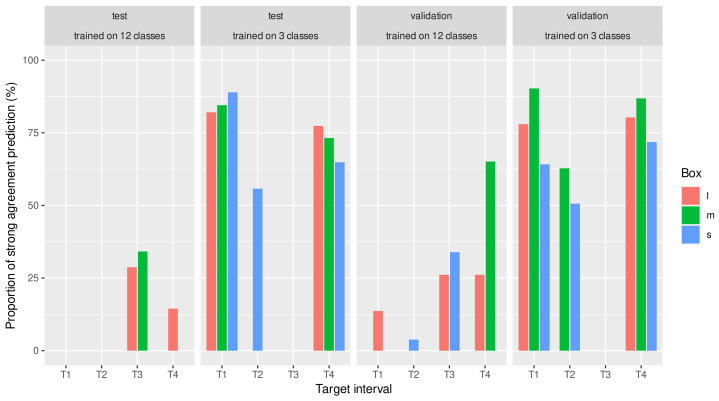
Proportion of predictions with strong agreement (Cohen’s kappa ≥0.8). For predictions over different target ranges (T1–T4). CNNs trained in three BCS classes showed a remarkably higher proportion of strong agreement than those trained in 12 classes and then reclassified.

**Table 1 animals-13-00194-t001:** The number of annotated images included in the study per score. Images from sites F1 and F2 were used to create the training and validation sets, while the held-out test set only consisted of images from the F3 site.

BCS	F1 & F2 Farm		F3 Farm
	Training	Validation	Test
1.00	162	41	22
1.50	215	54	31
2.00	363	91	53
2.50	370	93	65
2.75	182	46	42
3.00	323	81	41
3.25	381	95	50
3.50	659	165	66
3.75	74	18	9
4.00	58	14	11
4.50	46	12	6
5.00	89	22	11
Total:	2922	732	407

**Table 2 animals-13-00194-t002:** Model selection. Ten pre-trained models of Detectron2 were run on the same data set with the same settings (number of epochs: 15). The model with the lowest validation loss was chosen for all further experiments.

Pretrained	Validation
Faster R-CNN Model	Loss
R_50_FPN_3x	0.0612
R_101_FPN_3x	0.0628
R_50_FPN_1x	0.0637
X_101_32x8d_FPN_3x	0.0662
R_50_DC5_1x	0.0796
R_50_DC5_3x	0.0840
R_101_C4_3x	0.0848
R_101_DC5_3x	0.0848
R_50_C4_1x	0.1019
R_50_C4_3x	0.1040

**Table 3 animals-13-00194-t003:** Body condition score target intervals: T1 for calving (DIM: 0) and the dry period (DIM from −60 to −1), and dry off period (DIM > 300 and DIM < −60); T2 for early (DIM: 1–30) and mid-lactation (DIM: 101–200), T3 for peak lactation (DIM: 31–100) and T4 for late lactation (DIM: 201–300). Below, the relabeling for each interval is shown. For each regime, the original ordinal labels were relabeled according to the given threshold of that regime. For example, with the T1 thresholding, there were more images with cows in the “below” class, whereas if we relabeled the data with the T2 thresholding, then the three classes were more evenly split. Class distributions in the training and validation sets match respective test sets.

Mark	Target BCS Interval	
	Min	Max
T1	3.25	3.75
T2	2.75	3.25
T3	2.50	3.00
T4	3.00	3.75
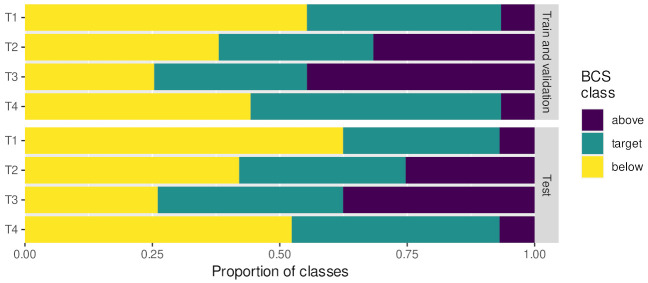		

## Data Availability

The weights used for the predictions from training with each BCS class and annotation box combination can be downloaded from https://doi.org/10.6084/m9.figshare.21372000.v1 (accessed on 28 December 2022).

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
