# Peer review of "Impact Evaluation of Score Classes and Annotation Regions in Deep Learning-Based Dairy Cow Body Condition Prediction"

_animals, 2023, doi:10.3390/ani13020194_

Round 1

Reviewer 1 Report

The subject of the presented article is the broader context of evaluating the condition of dairy cows using the classification of their fat reserves, generally known and referred to as BCS. This method is effective and very friendly to animals. There is a sufficient number of studies that prove the connection between inadequate condition and problems in the production or reproduction of dairy cows.

I consider the large number of animals included in the experiment (1150, 880 and 960 cows, apparently of the Holstein breed) to be the indisputable merit of the presented study.

A short introduction on whether an experiment carried out on a breeder with a different utilitarian focus would yield the same results.

Minor formal deficiencies can be 0corrected during the final proofreading.

I recommend accepting the submitted work for printing.

Author Response

Referee:
A short introduction on whether an experiment carried out on a breeder with a different utilitarian focus would yield the same results.

Answer:
Thank you for your suggestion, we have added the following to the discussion. "The presented results suggest that similar outcomes can be expected in condition scoring for other breeds and types of utilization. But, this assumes that similar CNNs should be trained on datasets generated by a scoring system applied to given breeds and types of utilization. Thus, the trained algorithm presented here cannot be used one-to-one for other breeds and utilisation types."

Reviewer 2 Report

Dear authors,

My concerns below are mostly moot suggestions but there are a few wording issues that may mislead the reader.

The abstract is brief, but you may be within a constraint...
8     we obtained a minimum or WEAK agreement
17+   stores -> deposits
17    supply -> surplus, as we are dealing with energy balances here.
18    saturation of fat and energy stores -> accumulation of ... stores ?  saturation is really misleading when you refere to fat  (vs double, triple bonds).
23    condition points -> scores (as in the acronym BCS)
23    abnormalities -> conditions? health issues? mismanagement?
25     wrong closing parenthesis
26    a consequence of lameness -> a cause for ?
32    , -> and
41    four different target intervals (please include here a reference to lactation stage, despite being fully explained later)
42    interest of region rectangles of the rump -> I would appreciate a reference to "region of interest" ROI, and to the practicalities of fixed mount cameras for production operation, etc
53    by an expert remotely -> could be you be meaning "deferred" or "at a later date" as in "not simultaneously"?
66    this long phrase is critical and probably requires better redaction
Fig 1    this box limits are critical and require more precise anatomical references
93    Let me suggest this wording: Following the"one-versus-all" scheme, accuracy was calculated
123    references to "dry" and "dry off" as different stages without further detail is a bit confusing
166    body scoring classes are ordinal but not continuous. -> Do you mean "not linear"? This is an interesting observation, is it yours or is there a reference?
Fig 2&3    missing y axis label and description
177    large boxes rank lower in each case; referred in Discussion (267) regarding information and noise but it may be done formally for great effect.
242    Incorporation of repeated  observations, a key issue that may require more discussion at this stage. 
255    inter-assessor agreement deserves a bit of review in the introduction as is one of the main Conclusions.
274    this observation I found in ancient sheep scoring advice. The mighty 13th century spanish breeders association La Mesta forbade touching for scoring to promote remote scoring. Still i would like to see the suggested ensemble checked... whether tactile or 3D input create richer training sets for the algorithm beyond what is picked by the eye of the expert.

Author Response

Referee:
The abstract is brief, but you may be within a constraint...

Answer: 
We have extended the abstract by a sentence, now, it contains 198 words under the limit of 200.

Referee:
8     we obtained a minimum or WEAK agreement

Answer: 
Thanks for the comment, we have edited it.

Referee:
17+   stores -> deposits

Answer: 
Thanks for the comment, we have edited it.

Referee:
17    supply -> surplus, as we are dealing with energy balances here.

Answer: 
Thank you for your comment, but the authors feel that 'supply' is a better description, as the BCS scale used is also suitable for assessing lean animals, but not for 'energy surplus'.

Referee:
18    saturation of fat and energy stores -> accumulation of ... stores ?  saturation is really misleading when you refere to fat  (vs double, triple bonds).

Answer: 
Thanks for the comment, we have edited it.

Referee:
23    condition points -> scores (as in the acronym BCS)

Answer: 
Thanks for the comment, we have edited it.

Referee:
23    abnormalities -> conditions? health issues? mismanagement?

Answer: 
Thanks for the comment, we have edited it.

Referee:
25     wrong closing parenthesis

Answer: 
In the brackets we only give an example of a metabolic problem, placental retention is not thought belonging to there.

Referee:
26    a consequence of lameness -> a cause for ?

Answer: 
Thanks for the comment, we have reworded the sentence.

Referee:
32    , -> and

Answer: 
Thanks for the comment, we have edited it.

Referee:
41    four different target intervals (please include here a reference to lactation stage, despite being fully explained later)

Answer: 
We have extended the sentence by the target intervals and DIM periods.

Referee:
42    interest of region rectangles of the rump -> I would appreciate a reference to "region of interest" ROI, and to the practicalities of fixed mount cameras for production operation, etc

Answer: 
Thanks for the comment, we have re-worded the sentence.

Referee:
53    by an expert remotely -> could be you be meaning "deferred" or "at a later date" as in "not simultaneously"?

Answer: 
Thanks for the comment, we have reworded the sentence.

Referee:
66    this long phrase is critical and probably requires better redaction

Answer: 
Thanks for the comment, we have divided and reworded the sentence.

Referee:
Fig 1    this box limits are critical and require more precise anatomical references

Answer: 
Thanks for the comment, we have extended the description with more precise anatomical names.

Referee:
93    Let me suggest this wording: Following the"one-versus-all" scheme, accuracy was calculated

Answer: 
Thanks for the suggestion, we amended it.

Referee:
123    references to "dry" and "dry off" as different stages without further detail is a bit confusing

Answer: 
Thanks for the comments, we have clarified the stages, and it was reworded in the whole text. 

Referee:
166    body scoring classes are ordinal but not continuous. -> Do you mean "not linear"? This is an interesting observation, is it yours or is there a reference?

Answer: 
It was published by Song et al. (2019), and now we have inserted the reference into the sentence mentioned.

Referee:
Fig 2&3    missing y axis label and description

Answer: 
Thanks for the comment, we added the title for the y-axis of Fig 2, 3, 6.

Referee:
177    large boxes rank lower in each case; referred in Discussion (267) regarding information and noise but it may be done formally for great effect.

Answer: 
This is not quite how we see the results in the mentioned sentences: "Based on the test set, the prediction precision per box in descending order for the target ranges is as follows: T1: m, s, l; T2: s, m, l; T3: l, s, m; T4: m, l, s. In the validation set T1: m, l, s; T2: m, l, s; T3: s, l, m; T4: m, l, s."

Referee:
242    Incorporation of repeated  observations, a key issue that may require more discussion at this stage.

Answer: 
In training sets, the repeated (unbalanced) occurrence of the animals is a real issue, but we discuss the multiple predictions on all animals to be studied to improve the precision. 

Referee:
255    inter-assessor agreement deserves a bit of review in the introduction as is one of the main Conclusions.

Answer: 
Thanks for highlighting the leak, the sentence within the introduction: "The volatility of intraobserver and interobserver agreements makes the data generated challenging to use [7]." is extended by the following:
"Mullis et al.[8 ] show that the agreement between two experts’ BCS values is moderate, while Song et al.[9] found that an inter-assessor and intra-assessor agreement is weak and moderate, respectively."

Referee:
274    this observation I found in ancient sheep scoring advice. The mighty 13th century spanish breeders association La Mesta forbade touching for scoring to promote remote scoring. Still i would like to see the suggested ensemble checked... whether tactile or 3D input create richer training sets for the algorithm beyond what is picked by the eye of the expert.

Answer: 
That would be the subject of further studies. By the way, we would be very interested in the historical sources to improve our teaching materials. 

Reviewer 3 Report

Dear authors,

As a bovine healt diplomate, I have read your concept paper with interest and to advice the Editor to accept, your paper needs some improvement, especially the discussion. The goal of the discussion is to discuss the results of your study with the literature and that is minimal now

in detail:

abstract: replace week by weak

page 4, table 2: 4 figures behind the dot is too much here, 2 is sufficient

page 7 at the bottom: what means m,s, l or 1, please give some explanation here, you losing not only me here

page 9: Discussion, see above, 2nd paragraph, starting with However, ... approach can be removed 

3rd paragraph: here you have also look, how other researchers have done this, you can skip until page 10 The use of ....

starting with The use of..., that paragraph is OK, but a paragraph later and the other and the other not any comparison with the literature, this is not acceptable in this stadium 

So, rewrite the Discussion. 

Author Response

Referee:
abstract: replace week by weak

Answer: 
Thanks for the comment, we have edited it.

Referee:
page 4, table 2: 4 figures behind the dot is too much here, 2 is sufficient

Answer: 
At first glance, the 4 decimal places may indeed be too many, but in the case of the proposed 2, the order and model choice presented cannot be followed. If the Referee has a proposal for model selection based on two decimal places, we would be glad to consider its applicability.

Referee:
page 7 at the bottom: what means m,s, l or 1, please give some explanation here, you losing not only me here

Answer: 
We agree with the referee, those letters for the readers are important to understand our work. Without the meaning of those symbols, the reader can't catch the paper. Since those letters are used in Figure 1-7 and in the results, without the interpretation of them is hard to imagine some are able to perceive the manuscript. To help the readers the abbreviations were introduced in materials and methods.  

Referee:
page 9: Discussion, see above, 2nd paragraph, starting with However, ... approach can be removed 
3rd paragraph: here you have also look, how other researchers have done this, you can skip until page 10 The use of ....

Answer: 
It is unclear what the referee is writing about, as there is no reference to other authors in that section. This and your previous suggestion mean that nearly half of the discussion should be removed. Since neither the other two reviewers nor the editor has suggested a similar change, please explain exactly which sentences you consider inappropriate and why.

Referee:
starting with The use of..., that paragraph is OK, but a paragraph later and the other and the other not any comparison with the literature, this is not acceptable in this stadium

Answer: 
In these sections, comparisons are missing because similar work has not been done by others. If you know of such work, please let us know, and we will include it in the discussion.

Round 2

Reviewer 3 Report

Dear authors,

Thank you for re-submitting the paper to the Journal and I have seen that you seriously tried to improve the manuscript and you succeeded. But I still have serious problems with the discussion, which has the objective to discuss the outcome of your study with the literature. You have to improve that part before I can advise the Editor to accept.

minor points:

page 1. explain the abbreviations (R-CNN, CNN) in the abstract, otherwise you lose your readers already here.

line 43, skip Too low condition... . is and is Bewley and Schutz a correct ref. here?? Next sentence: ... with a too high decrease ...

Re-evaluate your manucript again before I can evaluate (sorry)

Author Response

Referee: Thank you for re-submitting the paper to the Journal and I have seen that you seriously tried to improve the manuscript and you succeeded. But I still have serious problems with the discussion, which has the objective to discuss the outcome of your study with the literature. You have to improve that part before I can advise the Editor to accept.

Authors: In the previous round, we also asked the referee to help us improve the discourse beyond his general comments and radical deletion suggestions. Unfortunately, his current comment does not inform us where, why, and how the manuscript should be amended.  
Since we could not find any substantive suggestions, we thought we could find some orientation points in the assessment given in the Review Report Form. But we only see that Referee suggests improvements in one area: "Is the research design appropriate? (Can be improved)". In contrast, the Referee does not mention this in its detailed review. This is interesting, as we have made some clarifications in the methodology section, considering the suggestions of the other two reviewers. Before these changes, Referee 3's assessment on the same point was not "Can be improved" but "Yes". We would like to continue to ask the referee to help us improve our manuscript with clarified advice.

Referee: page 1. explain the abbreviations (R-CNN, CNN) in the abstract, otherwise you lose your readers already here.

Answer: The readability of the abstract has been improved with the necessary words.  

Referee: line 43, skip Too low condition... . 

Answer: In the original manuscript version, the candidate sentence "Too low ..." was line 26 in the previous version, which has been reworded as requested by Referee 2 in its current form. We would like to ask the opinion of Referee 2 before deleting it. 

Referee: is and is Bewley and Schutz a correct ref. here??

Answer: We checked, and as with the Referee's assessment (in the Review Report Form: "Are all the cited references relevant to the research? Yes"), we came to the same conclusion. For the relevant topics, some copied sentences from the cited paper:

Reduced milk production:
"For example, in early British work, cows calving with BCS <2 (US BCS <2.75) produced below their potential milk yield whereas those calving with BCS above 2.5 (US BCS >3.25) produced above their potential milk yield."

Lameness:
"In a German study, cows with BCS <3.0 at calving and during early lactation were more likely to be lame."

Metritis:
"Metritis is an inflammation of the lining of the uterus, most prevalent during early lactation. Butler and Smith (1989) reported significantly higher incidence of metritis in cows losing 0.5 to 1.0 BCS units (22%) or >1.0 BCS units (47%) when compared with cows losing <0.5 BCS units (6%). Markusfeld et al. (1997) demonstrated that cows losing more BCS during the dry period were more likely to experience metritis. Titterton and Weaver (1999) observed higher uterine discharge scores for cows calving with BCS ≤2.5 (≤3.25 US BCS) or ≥3.5 (≥4.25 US BCS) than for cows calving with BCS of 3.0 using the Mulvany (1977) BCS system. Heuer et al. (1999) computed a higher odds ratio (1.9) among thin cows (BCS ≤2.0) as compared with normal or fat cows. Kim and Suh (2003) observed significantly higher incidence of metritis in cows losing ≥1 point of BCS between dry-off and “near calving” than in cows that lost <1 point during the dry period. In a German study, cows with BCS at calving <3.0 were more likely to have metritis than cows with a higher BCS at calving (odds ratio = 2.95; Hoedemaker et al., 2008). Waltner et al. (1993) failed to identify a relationship between BCS and metritis."

Displaced abomasum:
"Hoedemaker et al. (2008) reported that cows with higher BCS losses during early lactation were more likely to have a displaced abomasum."

Open days:
"Fagan et al. (1989) demonstrated that cows with a BCS <2.5 had longer calving intervals than those with BCS ≥2.5."

Inactive ovaries:
"In the same study, cows that lost more BCS during the dry period were 2.1 times more likely to have inactive ovaries for each additional BCS unit lost. Opsomer et al. (2000) calculated odds ratios for delayed ovarian function of 18.7 and 10.9 for cows losing more body condition during the first and second months of lactation, respectively."

Referee: Next sentence: ... with a too high decrease ...

Answer: We have tried to understand the requirement of the Referee, but we need some more help. Basically, we don't understand how any decline can be too high.